# Treatment of Malignant Adnexal Tumors of the Skin: A 12-Year Perspective

**DOI:** 10.3390/cancers14040998

**Published:** 2022-02-16

**Authors:** Marcin Kleibert, Iga Płachta, Anna M. Czarnecka, Mateusz J. Spałek, Anna Szumera-Ciećkiewicz, Piotr Rutkowski

**Affiliations:** 1Department of Soft Tissue/Bone Sarcoma and Melanoma, Maria Sklodowska-Curie National Research, Institute of Oncology, 02-781 Warsaw, Poland; marcin.kleibert@interia.pl (M.K.); iplachta@outlook.com (I.P.); mateusz@spalek.co (M.J.S.); piotr.rutkowski@pib-nio.pl (P.R.); 2Faculty of Medicine, Medical University of Warsaw, 02-091 Warsaw, Poland; 3Department of Pathology and Laboratory Diagnostics, Maria Sklodowska-Curie National Research Institute of Oncology, 02-781 Warsaw, Poland or anna.szumera-cieckiewicz@pib-nio.pl; 4Department of Diagnostic Hematology, Institute of Hematology and Transfusion Medicine, 00-791 Warsaw, Poland

**Keywords:** radiotherapy, rare skin cancer, sebaceous, follicular, apocrine, eccrine, rare neoplasm, skin neoplasm, treatment, adnexal carcinoma

## Abstract

**Simple Summary:**

Surgical resection is the standard of care for malignant adnexal cancers of the skin. The efficacy of radiotherapy and targeted therapies is still undetermined in both adjuvant and palliative settings. In routine clinical practice, genetic abnormalities are rarely assessed, although numerous mutations have been described, e.g., in *TP53* (tumor protein p53), *FAT2* (FAT atypical cadherin 2), *KMT2D* (lysine methyltransferase 2D), *CACNA1S* (calcium voltage-gated channel subunit alpha1 S), and *PTEN* (phosphatase and tensin homolog). These cancers occur mainly in the middle-aged, and are locally aggressive, but when metastasized result in poor prognosis. The aim of this study is to analyze treatment efficacy in patients treated at a reference center since 2009. The role of multidisciplinary treatment is also discussed.

**Abstract:**

Malignant adnexal cancers of the skin—extremely rare neoplasms—are mostly reported as non-symptomatic, slow-growing nodules. These carcinomas occur mainly in the middle-aged (50–60 years of age); they are mostly localized on the upper part of the body and are locally aggressive, infiltrate surrounding tissue, and metastasize to regional lymph nodes. The patients’ outcomes depend on multiple prognostic factors, including the size of the primary tumor and its mitotic count. Surgical resection of the primary tumor with or without regional lymph nodes is the treatment method of choice; however, due to aggressive tumor behavior, perioperative treatment may be considered. The role and efficacy of radiotherapy in the treatment of skin adnexal malignancies are not yet fully defined. Some authors suggest that adjuvant radiotherapy may be considered in locally advanced and regional disease. The aim of this study was to evaluate treatment outcomes and assess the efficacy of combined therapy in patients with adnexal malignancies. Our analysis covered all cases of cutaneous adnexal tumor patients diagnosed and provided with multidisciplinary treatment with surgery and radiotherapy since the beginning of 2009.

## 1. Introduction

Malignant adnexal tumors of the skin (MATS) are an infrequent, heterogeneous group of tumors. MATS originate from eccrine, apocrine, and sebaceous sweat glands (Figure 1). Adnexal carcinomas remain a rare finding, but their incidence is increasing—partially in association with the aging of the population [1,2,3]. Proper diagnosis may be challenging even for experienced pathologists. Surgical removal of the primary tumor with or without regional lymph nodes is the treatment of choice; however, perioperative radio- or chemotherapy should be considered in the event of aggressive behavior. In fact, the optimal treatment approach for MATS patients is not yet established, as we discussed in our recent reviews [4,5]. Thus, analysis of more cases is necessary in order to provide additional clinically relevant data. The malignant potential of MATS is mainly locoregional, and it is challenging to perform radical surgery microscopically; therefore, neoadjuvant or adjuvant treatment—particularly radiotherapy—should always be considered as a crucial part of the overall therapeutic approach to improve local control, and should be discussed by a multidisciplinary tumor board (MTB). The use of radiotherapy may also be beneficial in treating lymph node metastases, which are common among patients with MATS [6]. However, acute and particularly late toxicities need to be taken into account in the decision process. In our case series, we aimed to determine the efficacy of multidisciplinary therapy in subsequent MATS patients treated at a national reference skin cancer department.

The aim of this study was to evaluate treatment outcomes and assess the efficacy of combined therapy in patients with cutaneous adnexal tumors treated at a tertiary skin cancer center.

## 2. Materials and Methods

We performed a retrospective cohort analysis of all MATS diagnosed at our center between 2009 and 2021. The inclusion criteria were as follows: >18 years of age, pathological diagnosis of non-Merkel-cell MATS, radiotherapy as an integral mode of treatment, and available complete electronic medical records. The following data were analyzed: demographic data, pathological classification, primary site of the disease, multidisciplinary tumor board treatment plan, type of surgical treatment, indication for radiotherapy, radiotherapy dose and technique, chemotherapy, local disease control, and stage of the disease, including the incidence of lymph node metastases and distant metastases.

## 3. Results

### 3.1. Tumors with Follicular Differentiation

#### 3.1.1. Trichilemmal Carcinoma

An 81-year-old woman presented with a recurrent skin lesion on the right palm consisting of a few nodules infiltrating the postoperative scar. The primary lesion was surgically excised one month earlier in another hospital, and a diagnosis of trichilemmal carcinoma was made (Figure 2(A1,A2)). Because the patient refused forearm amputation and enlarged right axillary lymph node packet, chemotherapy consisting of paclitaxel (60 mg in a dose) and cisplatin (20 mg in a dose) was administered. During the second administration of paclitaxel, allergic response manifesting as fainting with preserved cardiac function and indeterminate blood pressure was observed; therefore, radical radiotherapy of a total dose of 20 Gy was applied in five fractions on an area of the palm and antebrachium. Subsequently, cisplatin monotherapy was offered; however, after the first administration, the therapy was ceased because of significant progression of the skin lesions of the right upper extremity and axilla despite the administered treatment. The patient was referred to hospice care.

#### 3.1.2. Pilomatrix Carcinoma

A 38-year-old woman presented with a small, unmovable nodule on the left side of the neck in the area of the submandibular salivary gland, which had appeared four months earlier. The fine-needle biopsy revealed numerous epithelial cells with signs of keratosis and atypia during the first consultation. The differential diagnosis between a ruptured epithelial cyst and squamous-cell carcinoma consisted of repeat biopsy and PET-CT. An increased uptake of ^18^F-FDG in the area of the left submandibular gland was detected. The tests confirmed the diagnosis of neoplasm. The radical excision of the lesion was performed two months after the first consultation. The pilomatrix carcinoma was diagnosed histopathologically with the capsule of the neoplasm in the surgical margin (R1) (Figure 2C).

Due to the lack of radical excision, intensity-modulated radiation therapy was introduced; the patient received 66 Gy (2 Gy per fraction). The follow-up lasted five years (visit every six months), with imaging and lab tests listed in the table (Table 1).

#### 3.1.3. Malignant Proliferating Pilar Tumor

A 73-year old woman with a reddish, painless node up to 6 cm in the largest dimension on the right occipital area and small nodules in the left parietal area was consulted, and qualified for the resection of the bigger lesion. The histological examination revealed a malignant proliferating pilar tumor with a presence in the surgical margin (R1). The patient was referred for adjuvant radiotherapy by the multidisciplinary tumor board. The lesion (atheroma) from the parietal area was resected before the first radiation treatment. The patient received 20 Gy in total (4 Gy per fraction), and refused further treatment. During a 1.5-year follow-up, no recurrence was observed.

### 3.2. Tumors with Sebaceous Differentiation

#### 3.2.1. Sebaceous Carcinoma

##### First Case

A 54-year old patient was consulted due to confirmed metastases to the cervical lymph node 11 months after non-radical resection of the SCC from the parietal area. Physical examination revealed enlarged LNs in the left (1 cm) and right (1 and 1.5 cm) IIa and IIb cervical levels. The patient was referred for CT and PET-CT. Additionally, fiber-optic examination ruled out any lesions in the upper respiratory tract. In two weeks, the patient was qualified for radical radiotherapy (photon x 6 MeV, technique IMRT), and developed metastases in the parietal lymph nodes, as confirmed by PET-CT. The metastatic lymph nodes (parietal and II with III cervical levels) were radiated with 67.5 Gy in total (2.25 Gy per fraction; 60 Gy (2 per fraction) was applied to the surroundings (I–V cervical levels, as well as nuchal and parietal lymph nodes). The radiation was stopped due to the fast progression of the metastases and the necessity of updating the radiation plan. Complete clinical regression was observed during treatment. After seven months, the patient was referred for a fine-needle biopsy due to a suspected lymph node above the left mastoid, and consulted with a neurologist due to paresthesia in the left parietal area and positive Lhermitte’s sign. The resection of the lymph nodes and lesion on the neck was performed after detecting cancer cells in the biopsy. The histopathological examination revealed the relapse of cancer and the mistake that was made during the assessment of the first specimen. The sebaceous carcinoma was diagnosed (immunohistochemical profile: (CK7+, CK5/6 focally +)), and the patient was referred for US every two months and PET-CT (Figure 2E). The reoperation was performed due to a positive surgical margin. During 2 years of follow-up, the patient underwent one more lymphadenectomy (right cervical level III), and developed leukopenia and thrombopenia.

##### Second Case

A 66-year-old woman, previously diagnosed with endometrial carcinoma and colorectal carcinoma, presented with a lesion in the supraclavicular area, measuring 3–4 cm in diameter. Two satellite tumors around the main tumor were identified. The primary lesion was incompletely excised six months earlier, and was diagnosed as sebaceous carcinoma. Radical radiotherapy (margin: 4 cm) with a total dose of 60 Gy was applied in 30 fractions. Then, resection of the residual tumor and scar tissue was performed after the previous excision. However, a histopathological exam showed no neoplastic tissue. Due to extended wound healing, the biopsy was performed seven months after the procedure, and neoplastic cells were not detected again. In a one-year follow-up, no evidence of recurrence or progression was detected.

##### Third Case

A 55-year-old man presented with a tumor in the lumbar region. According to the patient’s observations, the lesion had been present for several years. The primary lesion was incompletely excised in another hospital, and a histopathological exam showed sebaceous carcinoma G3 (immunohistochemical profile: CKAE1/AE3 (+), CK7(+), CK20(−), D2-40(−), TTF1(−), p40(−), S100(−)). The residual lesion was re-excised, and preoperative radiotherapy delivering a total dose of 25 Gy in five fractions was also applied (Figure 3A). Seven months later, a palpable enlarged inguinal lymph node was identified, and a fine-needle biopsy revealed neoplastic cells. Dissection of the left obturator lymph nodes was performed, and postoperative radiotherapy (total dose of 56 Gy in 28 fractions) was also applied (Figure 3B). In a four-year follow-up, the patient remained without evidence of recurrence or metastasis.

##### Fourth Case

A 60-year-old man was referred to a tertiary referral level hospital due to recurrence of the tumor after three incomplete (12/2009, 02/2010, 05/2011) resections of the lesion on the head. Sebaceous carcinoma was diagnosed. The medical council decided to resection, with regional lymphadenectomy and adjuvant radiotherapy to the lymph node bed (66 Gy, 2 Gy per fraction). The presence of cancer cells in the surgical margin and several lymph nodes was revealed during the postoperative histopathological examination. The intensity-modulated radiation therapy was initiated three months after surgery. The patient was discharged in good condition; relapse was diagnosed 14 months later. Three additional resections were performed, and the patient was referred for palliative chemotherapy (cisplatin + 5-fluorouracil; after progression—Taxol; second progression—methotrexate). The further history of the patient is unknown due to their referral to hospice care.

### 3.3. Tumors with Apocrine and Eccrine Differentiation

#### 3.3.1. Eccrine Porocarcinoma

A 54-year-old presented with a mass of up to 6 cm in the largest dimension in the right supraclavicular area. The lesion was primarily excised at another hospital, and eccrine porocarcinoma was diagnosed (immunohistochemical profile: CK7(+), BerEP4(+), p63(+), EA(+), CK19(+ in ~30%), CD117(focally +), D240(−), S100(−)). Because of incomplete resection, reoperation was performed two months after primary treatment. An ^18^F-FDG PET scan revealed metabolically active lymph nodes in the right axillary area. Biopsy was performed twice, but neither revealed metastases in the axillary lymph nodes (Figure 2B). However, lymphadenectomy was performed, and metastases were identified in 5 of 24 excised lymph nodes. It was decided to perform adjuvant radiotherapy with a total dose of 60 Gy in 30 fractions. Five months later, pulmonary metastases were found via CT. The patient received a chemotherapy regimen combined with CAV (vincristine 2 mg, doxorubicin 50 mg, cyclophosphamide 1000 mg) and EP (etoposide 100 mg, cisplatin 20 mg), and then metastasectomy with a microscopically negative margin was performed. After a few months, new metastatic lesions were found in the lung, and chemotherapy with etoposide (dose: 200 mg) and cisplatin (dose: 40 mg) was performed. The patient currently continues the chemotherapy.

#### 3.3.2. Mucinous Carcinoma

A 70-year-old woman presented with a nodular bleeding lesion on the scalp, measuring approximately 8 cm in diameter, and the tumor was surgically excised. Histopathological examination showed cutaneous mucinous adenocarcinoma with positive surgical margins and the following immunohistochemical stains: ER(+), PGR(+), PAS (+) alcian blue (+), CK7(+), CK AE1/3(+), CK20(−), CEA(+), EMA(+), GCDFP-15(+/−), synaptophysin(+), chromogranin(−), CD56(−), TTF-1(−), and CK 5/6 with p63 (+) in cells surrounding the neoplastic nests. Fine-needle aspiration biopsy of the enlarged cervical lymph node (level V) was performed, and the result revealed metastasis. Lymphadenectomy of the right level V lymph nodes was therefore performed. Adjuvant radiotherapy was applied for tumor bed and locoregional lymph nodes (right cervical level II, III, IV, and V lymph nodes), with a total dose of 50 Gy (fraction dose: 2.5 Gy) and a radiation boost dose of 6 Gy. Six months after radiation therapy, nodules in the retroauricular and occipital regions emerged. Fine-needle aspiration biopsy showed neoplastic cells; therefore, incomplete (R1) resection was performed (Figure 2D). Six months later, another two retroauricular tumors arosewere also excised. Histopathological exam revealed perineural infiltration and vascular invasion with positive surgical margins. In addition, facial nerve paralysis occurred. Other nodules located in the retroauricular area were resected four months later. During the next appointment, multiple nodules in the retromandibular and retroauricular areas were observed, and the fine-needle biopsy revealed recurrence of the disease. Due to ineffective treatment and the patient’s condition, the patient was referred to receive palliative chemotherapy in another hospital.

## 4. Discussion

Cutaneous adnexal tumors are an extensive group of lesions exhibiting morphological differentiation toward one or more adnexal epithelial structures of the skin, including sebaceous glands, hair follicles, and apocrine or eccrine glands. Their etiology is unknown; however, UV and ionizing radiation, preexisting scars, xeroderma pigmentosa, Cowden syndrome, or solid organ transplantation have been reported to pose a risk of trichilemmal carcinoma development (Table 2) [1,7]. In our case of trichilemmal carcinoma, it is unknown whether the UV exposure was increased; nevertheless, the patient was neither exposed to long-term low-dose X-irradiation nor was she a transplant recipient. Porocarcinoma and pilomatrix carcinoma may arise from their benign counterparts; porocarcinoma may also arise due to prolonged radiotherapy [8,9,10]. In our cases, the patients did not observe previous benign lesions; furthermore, the patient with porocarcinoma did not undergo any radiotherapy. The association between pilomatrixoma and certain genetic disorders—such as Gardner syndrome, Rubinstein–Taybi syndrome, Sotos syndrome, myotonic dystrophy, and Turner syndrome—is well documented; however, it was not observed in pilomatrix carcinoma [11]. The risk of developing sebaceous carcinoma is increased by immunosuppression after solid organ transplantation; other immunosuppressed states, including lymphoma and previous radiotherapy, might cause predisposition to periocular sebaceous carcinoma [12]. In one of our cases of sebaceous carcinoma, four years before the diagnosis, the patient underwent brachytherapy (total dose: 12 Gy) to treat endometrial carcinoma; however, the sebaceous carcinoma was located in the supraclavicular area—not in the periocular, as is most common.

MATS are extremely rare neoplasms. Based on data from the U.S. Surveillance, Epidemiology, and End Program (SEER) of the National Cancer Institute, the age-standardized incidence rate (ASR) in the U.S. is 5.1 per 1 million person years, whereas in Europe ASR was reported to be 2.1 per million by Mallone et al. [2,22]. Therefore, a multidisciplinary treatment team of world-renowned experts in dermatologic oncology, dermatologic and Mohs surgery, dermatopathology, head and neck surgery, reconstructive surgery, radiation oncology, medical oncology, and radiology is required in order to optimize treatment that could relevantly affect patient outcomes. Lymph node metastases are common among patients with MATS. According to Robson et al., 47% of patients with apocrine carcinoma can develop metastases during follow-up (median, 2 years) [23]. Similar values were reported among other patients with carcinomas mentioned in this article [24]. Patients with additional regional lymph node involvement have lower median survival rates, and may benefit from lymphadenectomy and additional radiotherapy [25].

Sebaceous carcinoma is the most frequent cutaneous adnexal malignancy; this seems to be why four of the nine cases described in our study represented such a neoplasm. Lesions show a predilection for the head and neck, and half of our cases were located there: one in the occipital region and one in the parietal. Our patients were in their sixth and seventh decades of life (mean age: 58.8 years)—younger than patients in the studies included in a review by Owen et al. (mean age for extraocular lesions: 67.9 years; mean age for periocular lesions: 67.7 years) [12]. All lesions were extraocular. Our male-to-female ratio was 1:1. All of our patients underwent surgical excision; however, in all cases (4 of 4), primary treatment was non-radical (R1/R2 resection). Adjuvant radiotherapy was administered in all of our patients, with variable outcomes—in two patients to the area of the primary site and regional lymph nodes, in one case only to the primary site of the tumor, and in the last case only to the lymph nodes. In the follow-up, in two cases relapse was observed (after 14 months and after 7 months, respectively), whereas in another two follow-up (one- and four-year) was free of recurrence. In the literature, the mean time to recurrence was 19.4 (standard deviation: 20.4) months; in one of our cases, the relapse occurred 5.4 months faster. Moreover, most recurrences appear within 6 years of surgery, and none of our patients were observed for long enough to access the efficacy of the therapeutic approach. The role of radiation therapy remains unclear; recent guidelines recommend considering radiotherapy in the treatment of extraocular sebaceous carcinoma as an adjuvant treatment for lymph node metastases or recurrent disease, or as monotherapy for patients who cannot be operated on or who present with unresectable tumors. For periocular sebaceous carcinoma, adjuvant radiotherapy may be useful in tumors manifesting perineural invasion [12]. Because of the potential aggressive behavior and insufficient data regarding appropriate treatment strategies, continued prospective investigation is crucial in order to clarify the therapeutic management.

Adjuvant radiotherapy was included as the therapeutic approach in other described cases. Advanced radiation oncology techniques include external beam radiation therapy (EBRT), intensity-modulated radiation therapy (IMRT), and surface-mold computer-optimized high-dose-rate brachytherapy (SMBT). Some previously published reports have suggested that the use of radiation is recommended in the treatment of masses exceeding 5 cm; however, this has not been prospectively evaluated [26]. In our case of pilomatrix carcinoma with positive surgical margins, this provided adequate local tumor control, because in the 5-year follow-up neither recurrence nor metastases were found, providing further evidence that radiotherapy may be used as an adjuvant treatment in this type of tumor; it can decrease the risk of local recurrence—which is quite high among patients primarily diagnosed with pilomatrix carcinoma (~83%, an average of 11.6 months)—even among cases with tumor-free surgical margins, because of which long-term follow-up is recommended. Additionally, radiotherapy may be considered for patients with pilomatrix carcinoma when excision cannot be performed [27,28,29]. The patient with a proliferating pilar tumor received a decreased total dose; however, no recurrences or metastases were found in a 1.5-year follow-up. Limited data do not confirm the role of adjuvant therapy for this entity [30,31,32]; however, around 40 Gy was administered in most cases [4]. Long-term follow up is recommended due to the potential for recurrence even 10 years after excision [4]. In the case of trichilemmal carcinoma, radiotherapy was administered because of intolerance to chemotherapy, and no spectacular effect was observed. The overall 5-year survival rate of the patients diagnosed with trichilemmal carcinoma was 89.2%, as reported by Zhuang et al., and the presence of lymph node metastases along with the status of the surgery margins were found to be the key prognostic factors [33]. Our patient presented with fast local relapse after the R1 primary resection; therefore, the prognosis was unfavorable. The role of radiotherapy in the management of trichilemmal carcinoma has not yet been established, but in some cases it may be useful—especially among patients with incomplete resection [34,35]. Radiation therapy of porocarcinoma and mucinous carcinoma remains questionable, and in our cases did not provide adequate disease control. The death rate was 67% among patients with porocarcinoma and regional lymph node involvement. Salih et al. proved that adjuvant treatment did not significantly improve the prognosis (*p* = 0.458) or the overall survival (*p* = 0.790) in porocarcinoma [36]. Recent data do not substantiate the use of radiotherapy, as mucinous carcinoma does not tend to respond to radiation treatment [9,37,38].

Some mutations can cause predisposition to the development of the tumors described above (Figure 4). None of our patients underwent molecular testing. Currently, there are two avenues where molecular tests have become a part of standard patient management: diagnosis of hereditary cancer syndromes, and personalized treatment selection based on molecular characteristics of neoplasm tissues [39]. Appendageal tumors may be the manifestation of inherited conditions characterized by an increased probability of neoplastic lesions occurring. For instance, Brooke–Spiegler syndrome, caused by mutations within *CYLD*—a tumor-suppressor gene regulating NF-κB activation—is an autosomal-dominant disorder characterized by multiple benign cutaneous adnexal tumors (most commonly spiradenomas, cylindromas, and trichoepitheliomas [40]. Cowden syndrome—one of the disorders that have been linked to germline mutations in the phosphatase and tensin homolog (*PTEN*) gene—causes elevated risk of malignancies, including breast, thyroid, or endometrial cancer [41]. The typical cutaneous manifestation is multiple trichilemmomas, considered as the benign counterpart of trichilemmal carcinoma. Whereas transformation of the trichilemmoma to trichilemmal carcinoma remains questionable, O’Hare et al. reported a case of a patient with Cowden’s disease who developed trichilemmal carcinoma [13]. It is crucial to identify genetic disorders in patients with cutaneous tumors, because while trichilemmal carcinoma remains the neoplasm with the most favorable prognosis, the risk of internal malignancies is increased, and prophylactic screening tests may be helpful to diagnose these neoplasms earlier [33]. Similarly, screening for Muir–Torre syndrome—a variant of Lynch syndrome—remains a crucial diagnostic step. Muir–Torre syndrome, an autosomal-dominant condition caused by germline variants in the DNA mismatch repair genes, is characterized by the occurrence of at least one cutaneous sebaceous tumor associated with at least one visceral malignancy [42]. Individuals might be selected for genetic testing using the Mayo risk score, whereas patients aged <50 years with extraocular sebaceous carcinoma may be considered for tumor tissue mismatch repair protein immunohistochemistry testing [43]. However, immunohistochemistry testing remains moderately sensitive, and is not specific for reliable detection of Muir–Torre syndrome compared to genetic testing initiated based on clinical criteria [12]. Recently, Denisova et al. performed the analysis of WES (whole-exome sequencing) data that revealed that *TP53* (tumor protein p53), *FAT2* (FAT atypical cadherin 2), *KMT2D* (lysine methyltransferase 2D), and *CACNA1S* (calcium voltage-gated channel subunit alpha1 S) were the most frequently mutated genes among 14 cases of eccrine porocarcinomas [44].

In the event that any molecular testing was not performed, no targeted therapy was utilized. For the neoplasms diagnosed in our cases, several mutations were used as the targets for immunotherapy. For sebaceous carcinoma, tumor sequencing revealed RAR-β, androgen receptor, mTOR, and EGFR as the potential investigational approaches [45,46,47]. PD-1 inhibitors, approved by the U.S. Food and Drug Administration for microsatellite-unstable malignancies deriving from any tissues, may also be useful in therapy. Domingo-Musibay et al. described a case of a man with widely disseminated sebaceous carcinoma who achieved a near-complete durable response to pembrolizumab [48]. Kodali et al. reported a case of a woman who presented with rapidly progressing periocular sebaceous carcinoma, which responded to chemoimmunotherapy with pembrolizumab and carboplatin [49]. However, in both cases, microsatellite instability was not detected [50]. Other reported cases showed no response to anti-HGF antibody and nivolumab treatment [51]. Further studies are needed in order to assess the possible benefits of immunotherapy in the treatment of sebaceous carcinoma.

Immunotherapy may also be applied as a treatment of other adnexal malignancies. Lee et al. reported a case of a patient with metastatic porocarcinoma who achieved a clinical and radiological response when treated with pembrolizumab [52]. For hidradenocarcinoma, apocrine carcinoma, and signet-ring call/histiocytoid carcinoma, HER2 inhibitors or selective estrogen receptor modulators may be administered [53,54,55,56,57,58]. Additionally, anti-androgenic therapy is beneficial for patients with signet-ring call/histiocytoid carcinoma [59]. PI3K/Akt/mTOR pathway inhibitors were successfully applied in the therapy of apocrine carcinoma and proliferating trichilemmal tumors [30,60]. Another treatment was reported for trichoblastic carcinoma/carcinosarcoma. Lepesant et al. administered vismodegib with remarkable response in metastatic trichoblastic carcinoma [61]. Battistella et al. administered sunitinib to two patients with metastatic hidradenocarcinoma and trichoblastic carcinoma. In the case of hidradenocarcinoma, tumor stabilization with no additional metastases was achieved for over eight months before the patient relapsed. The patient with trichoblastic carcinoma achieved partial remission, and the disease stabilized after 10 months [62]. Currently, new treatment strategies are being developed; a phase II trial study is being conducted to access the efficacy of the talimogene laherparepvec and nivolumab in therapy of malignant sweat gland tumors, sebaceous carcinoma, and trichilemmal carcinoma (NCT02978625); additionally, a phase I multiple-dose study of XmAb^®^20717 is underway to evaluate the safety and tolerability in subjects with selected advanced solid tumors, including malignant adnexal neoplasms (NCT03517488).

## 5. Conclusions

MATS are a highly heterogenic group of cutaneous lesions. Due to the tendency towards locoregional and distant metastases, patients should be evaluated by MTB in reference skin cancer centers—especially in cases presenting clinical features of local aggressiveness. It is generally accepted that radical surgery should not be proposed for these cases if feasible. The efficacy of radiotherapy in adnexal tumors requires further investigation. However, we proved that radiotherapy can be beneficial—especially among patients with sebaceous, trichilemmal, and pilomatrix carcinomas, or malignant proliferating pilar tumors. Multi-institutional studies should be helpful to establish optimal management sequences—especially for cases with lymph node metastases and those after non-radical resection. At this point in time, it may be suggested that radiotherapy could significantly influence patients’ outcomes, and should be considered by MDT.

## Figures and Tables

**Figure 1 cancers-14-00998-f001:**
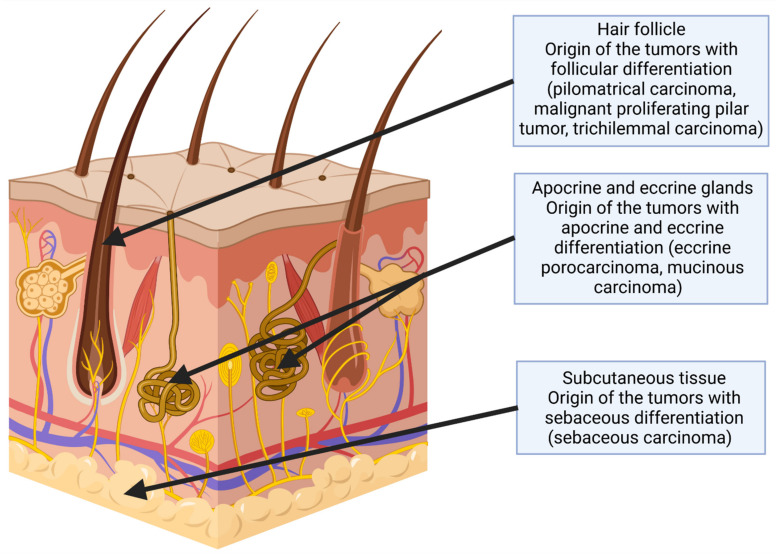
Skin specimen section: The arrows point to the tissues and structures that can be the origin of the appendageal tumors. The cancers described in this article are listed in brackets. Created with BioRender.com (18 October 2021).

**Figure 2 cancers-14-00998-f002:**
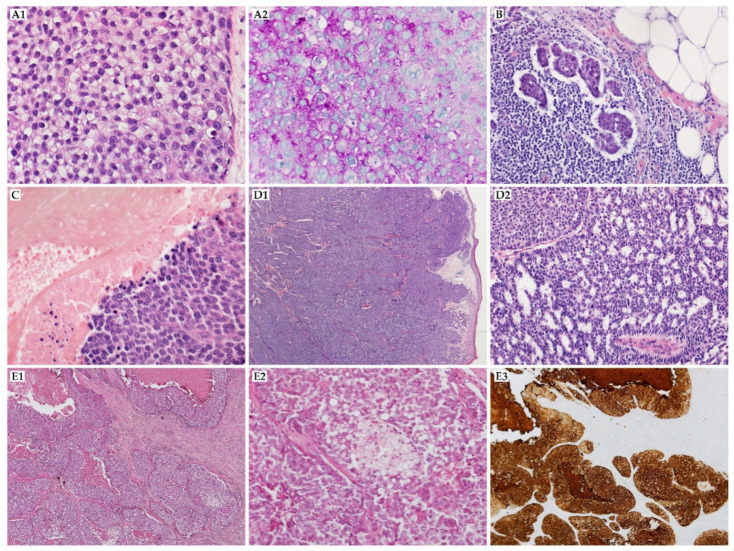
(**A**) Trichilemmal carcinoma (**A1**) (HE, 400×), characterized by lobules composed of large, atypical cells with clear cytoplasm and short cell borders; (**A2**) periodic acid–Schiff (PAS, 400×)-positive cytoplasmic reaction is used in differential diagnosis. (**B**) Lymph node metastasis of adnexal carcinoma (**B**) (HE, 200×) revealed malignant ductal and glandular structures; the differential diagnosis requires exclusion of the adenocarcinomas from other solid organs; previous or simultaneous involvement of the skin supports the diagnosis of skin adnexal carcinoma metastasis. (**C**) Pilomatrix carcinoma (**C**) (HE, 200×) with solid nests of immature, basaloid cells invading the deep dermis; in the right upper corner the shadow cells are visible, which are characteristic of the diagnosis. (**D**) Mucinous carcinoma: (**D1**) (HE, 20×) monotonous population of atypical cells with mucin seen in the borders; (**D2**) (HE, 100×) the in situ component is detected in most of the cases, and is essential for the diagnosis of the primary cutaneous neoplasms. (**E**) Sebaceous carcinoma: (**E1**) (HE, 20×) lobules of atypical cells with abundant necrosis; (**E2**) (HE, 200×) multivesicular clearing of cytoplasm should be identified in classical cases of sebaceous carcinoma]; (**E3**) tumor cells are EMA-positive; additionally, staining may improve visualization of the cytoplasm vacuolization (EMA, 20×).

**Figure 3 cancers-14-00998-f003:**
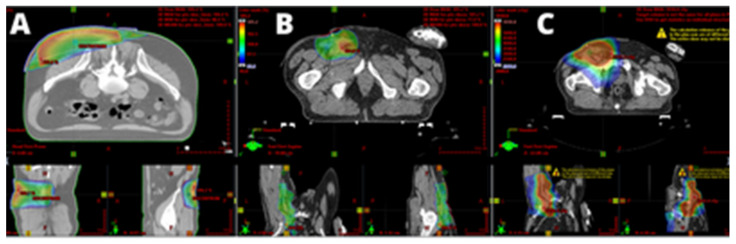
High-grade (G3) sebaceous carcinoma of the left lumbar area: On 24.11.2015, this patient received preoperative radiotherapy consisting of 25 Gy in five fractions (**A**), and underwent resection. After several months (5 July 2016), clinically palpable locoregional left inguinal lymph node metastases were present. Left ilioinguinal lymphadenectomy was performed, followed by postoperative radiotherapy consisting of 56 Gy in 28 fractions (**B**). The fusion of two radiotherapy plans was carried out in order to assess the tolerance of the proximal organs at risk (**C**).

**Figure 4 cancers-14-00998-f004:**
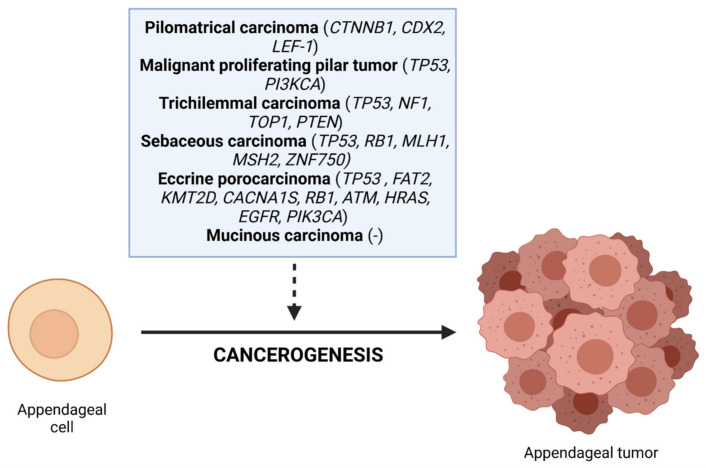
The characteristic mutations of tumors developed in the described patients. Created with BioRender.com (18 October 2021).

**Table 1 cancers-14-00998-t001:** Five-year follow-up of the patient with pilomatrix carcinoma. CBC: complete blood count; CT: computed tomography; US: ultrasound.

Time after the Surgery (Months)	Follow-Up
6	US of the cervical lymph nodes
12	Chest X-ray, head and neck CT, CBC, thyroid hormone levels
18	Chest X-ray, US of cervical lymph nodes
24	Clinical evaluation
30	Chest X-ray, head and neck CT
36	Abdomen US
42	Chest X-ray, US of cervical lymph nodes
48	Clinical evaluation
54	Chest X-ray
60	Head, neck, and abdomen CT
Time after the surgery (months)	Follow-up

**Table 2 cancers-14-00998-t002:** The syndromes and conditions related to higher risk of tumor development.

Tumor	Syndromes/Conditions that Potentially Increase the Risk	References
Trichilemmal carcinoma	UV and radiation exposure, pre-existing scars, Cowden syndrome, and immunosuppression	[13,14]
Pilomatrix carcinoma	UV exposure	[15]
Malignant proliferating pilar tumor	UV exposure	[16]
Sebaceous carcinoma	Muir–Torre syndrome, UV exposure, and immunosuppression	[17,18,19]
Eccrine porocarcinoma	Trauma, burning, radiotherapy, and immunosuppression, UV exposure, and AIDS	[20]
Mucinous carcinoma	UV and radiation exposure	[21]

## Data Availability

Raw pathology data were generated at the Department of Pathology and Laboratory Diagnostics, Maria Sklodowska Curie National Research Institute of Oncology, and are available from the authors on request.

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
