# Peer review of "Treatment of Malignant Adnexal Tumors of the Skin: A 12-Year Perspective"

_cancers, 2022, doi:10.3390/cancers14040998_

Round 1

Reviewer 1 Report

Despite the low number of patients, the authors should includes their results in the discussion and they should perform a comparison with the literature data

What did they want to prove?

Author Response

Thank you for the remark. We added the comparison of our results with literature data in the discussion. We wanted to present few more cases of these rare entities and to compare our treatment outcomes with data reported by other authors, to assess the usefulness of the radiotherapy in adnexal malignancies’ therapy as we have added in conclusions (416-419 lines). Additionally, the aim of the study was not clearly stated in the manuscript. Thus, we wrote “The aim of the study was to evaluate treatment outcomes and assess the efficacy of combined therapy in patients with cutaneous adnexal tumors treated in a tertiary skin-cancer center” (61-63 lines).

Reviewer 2 Report

Fig 2 -  the authors present images of histological preparations, but there are no indications of tissue changes. There is no use for pictures without a pathohistological description. 

Table 1 - table abbreviations are not clear

The manuscript needs to improve the description of pathohistology for all patients.

Author Response

Dear Reviewer, thank you for your important comments. Indeed, the histopathological descriptions seem to be important even if the publication is not only "histopathological oriented". We decided to improve the Figure 2 description and add the most important features, which may be interesting for Readers. We also added the missing abbreviations for table 1 and description of pathology results of our patients (staining).

Reviewer 3 Report

Congratulations to the authors for the interesting research, which provides therapeutic guidance for clinicians because adenxial tumors are quite rare cancers. Interestingly structured, well-written manuscript and supported with description of their pathogenesis. The presented cases are properly described.
The discussion section explains the case in the context of published information. The conclusions accurately and clearly explain the main clinical message.
The figures are of good quality and relevant to the clinical message.
The contribution of each author make to the case management and case report submission is clear.
The references are current.
Perhaps it would be worth supplementing the manuscript with a summary table: in which cases does a given tumor occur more often (e.g.
sebaceous carcinoma - Muir-Torre syndrome ...).

Author Response

Thank you for your kind comment. We added a table with risk factors for presented neoplasms.

Round 2

Reviewer 2 Report

Dear Authors,

adding description of pathohistological changes made the whole manuscript significantly better.

Please check: lines 313, 320 (space, spelling).